# User review analysis of dating apps based on text mining

**Qian Shen, Siteng Han, Yu Han, Xi Chen**  *

School of Statistics, Xi'an University of Finance and Economics, Xi'an, Shaanxi, China

* xchen0008@xaufe.edu.cn

## Abstract

With the continuous development of information technology, more and more people have become to use online dating apps, and the trend has been exacerbated by the COVID-19 pandemic in these years. However, there is a phenomenon that most of user reviews of mainstream dating apps are negative. To study this phenomenon, we have used topic model to mine negative reviews of mainstream dating apps, and constructed a two-stage machine learning model using data dimensionality reduction and text classification to classify user reviews of dating apps. The research results show that: firstly, the reasons for the current negative reviews of dating apps are mainly concentrated in the charging mechanism, fake accounts, subscription and advertising push mechanism and matching mechanism in the apps, proposed corresponding improvement suggestions are proposed by us; secondly, using principal component analysis to reduce the dimensionality of the text vector, and then using XGBoost model to learn the low-dimensional data after oversampling, a better classification accuracy of user reviews can be obtained. We hope These findings can help dating apps operators to improve services and achieve sustainable business operations of their apps.

## 1 Introduction

Maybe the term 'online dating' sounded weird in the 1990s, but today we have become accustomed to it. Mobile phones are 'virtual bars' in people's pockets, allowing singles to socialize wherever they want. At least 200 million people worldwide use digital dating services every month, a study of Smith and Duggan [1] found that one in ten Americans has used online dating websites or mobile dating apps; sixty-six percent of online daters have met someone they know through dating websites or apps, and 23 percent have met spouses or long-term partners through these sites or apps. One of Statista's report [2] claimed that in 2020, there would be 44.2 million online dating service users in United States. The company's digital market outlook estimated that number will increase to 53.3 million by 2025. Due to the COVID-19 pandemic since 2020, many activities of people have shifted from offline to online. It has also led to a significant increase in the frequency of online dating app users using them. Chisom's research work [3] suggested that loneliness and boredom due to adhering to the stay at home policy in the age of COVID-19, there was a rapid increase of online dating apps especially on Tinder

**Data Availability Statement:** All the data are available on figshare.com (https://figshare.com/articles/dataset/Text_of_user_reviews_of_dating_apps/21895827).

**Funding:** The author(s) received no specific funding for this work.

**Competing interests:** NO authors have competing interests.

and had in so many ways. In other words, dating apps have very good market prospects at present.

However, a good market prospect also means that there will be cruel competition among enterprises behind it. For operators of dating apps, one of the key factors in keeping their apps stable against the competitions or gaining more market share is getting positive reviews from as many users as possible. In order to achieve this goal, operators of dating apps should analyze the reviews of users from Google Play and other channels in a timely manner, and mine the main opinions reflected in the user reviews as an important basis for formulating apps' improvement strategies. The study of Ye, Law and Gu [4] found significant relationship between online consumer reviews and hotel business performances. This conclusion can also be applied on apps. Noei, Zhang and Zou [5] claimed that for 77% of apps, taking into account the key content of user reviews when updating apps was significantly associated with an increase in ratings for newer versions of apps.

For user reviews of apps presented in a textual state, we believe that text mining models can be used to analyze these reviews. Some researchers such as M Lee, M Jeong and J Lee [6] have studied the impact of online user negative reviews on consumers' choice when booking a hotel through text mining. Latent Dirichlet Allocation (LDA) was proposed by Blei et al. [7]. Since then, topic models based on LDA have become one of the key research areas of text mining. LDA is very widely used in the commercial fields. For example, Wahyudi and Kusumaningrum [8] have used an LDA-based topic model to perform sentiment analysis on user reviews of online shopping malls in Indonesia in their study.

Most of the sentences that people speak every day contain some kinds of emotions, such as happiness, satisfaction, anger, etc. We tend to analyze the emotions of sentences according to our experience of language communication. Feldman [9] thought that sentiment analysis is the task of finding the opinions of authors about specific entities. Operators of dating apps usually collect user feelings and opinions through questionnaires or other surveys within the websites or apps. For many customers' opinions in the form of text collected in the surveys, it is obviously impossible for operators to use their own eyes and brains to watch and judge the emotional tendencies of the opinions one by one. Therefore, we believe that a feasible method is to first build a suitable model to fit the existing customer opinions that have been classified by sentiment tendency. In this way, the operators can then obtain the sentiment tendency of the newly collected customer opinions through batch analysis of the existing model, and conduct more in-depth analysis as needed.

At present, many machine learning and deep learning models can be used to analyze text sentiment which is processed by word segmentation. In the study of Abdulkadhar, Murugesan and Natarajan [10], LSA (Latent Semantic Analysis) was firstly used for feature selection of biomedical texts, then SVM (Support Vector Machines), SVR (Support Vactor Regression) and Adaboost were applied to the classification of biomedical texts. Their overall results show that AdaBoost performs better compared to two SVM classifiers. Sun et al. [11] proposed a text-information random forest model, which proposed a weighted voting mechanism to improve the quality of the decision tree in the traditional random forest for the problem that the quality of the traditional random forest is difficult to control, and it was proved that it can achieve better results in text classification. Aljedani, Alotaibi and Taileb [12] have explored the hierarchical multi-label classification problem in the context of Arabic and propose a hierarchical multi-label Arabic text classification (HMATC) model using machine learning methods. The results show that the proposed model was superior to all the models considered in the experiment in terms of computational cost, and its consumption cost is less than that of other evaluation models. Shah et al. [13] constructed a BBC news text classification model based on machine learning algorithms, and compared the performance of logistic regression, random

forest and K-nearest neighbor algorithms on datasets. The results show that logistic regression classifier with the TF-IDF Vectorizer feature attains the highest accuracy of 97% for the data set. Jang et al. [14] have proposed an attention-based Bi-LSTM+CNN hybrid model that takes advantage of LSTM and CNN and has an additional attention mechanism. Testing results on Internet Movie Database (IMDB) movie review data showed that the newly proposed model produces more accurate classification results, as well as higher recall and F1 scores, than single multilayer perceptron (MLP), CNN or LSTM models and hybrid models. Lu, Pan and Nie [15] have proposed a VGCN-BERT model that combines the capabilities of BERT with a lexical graph convolutional network (VGCN). In their experiments with several text classification datasets, their proposed method outperformed BERT and GCN alone and was more effective than previous studies reported.

However, in practice when the text contains many words or the numbers of texts are large, the word vector matrix will obtain higher dimensions after word segmentation processing. Therefore, we should consider reducing the dimensions of the word vector matrix first. The research of Vinodhini and Chandrasekaran [16] showed that dimensionality reduction using PCA (principal component analysis) can make text sentiment analysis more effective. LLE (Locally Linear Embedding) is a manifold learning algorithm that can achieve effective dimensionality reduction for high-dimensional data. He et al. [17] believed that LLE is very effective in dimensionality reduction of text data.

Currently, there are fewer text mining studies on user reviews of apps that people use every day, but this field has caught the attention of researchers [18]. Much of the research on dating apps now focuses on psychology and sociology, with minority of studies looking at dating apps from a business perspective. The study by Ranzini, Rosenbaum and Tybur [19] found that Dutch people are more likely to choose Dutch people as potential partners when using dating apps, while Dutch people with higher education are more likely to choose potential partners with higher education backgrounds when using dating apps. Tran et al. [20] found that users of dating apps had significantly higher odds of unhealthy weight-control behaviors than those who had not used dating apps. Rochat et al. [21] used cluster analysis to study the characteristics of Tinder users. The results show that Tinder users participating in the study could be reasonably divided into four groups, and the users of each group were different in gender, marital status, depression and usage patterns. Tomaszewska and Schuster [22] compared perceptions related to sexuality of dating app users and non-dating app users, namely their risky sexual scripts and sexual self-esteem, and their risky and sexually assertive behaviors. Results showed that dating app users had more risky sexual scripts and reported more risky sexual behaviors than non-dating app users. In addition, male dating app users had lower sexual self-esteem and were more accepting of sexual coercion than male non-dating app users. Lenton et al. [23] studied the relationship between social anxiety and depressive symptoms of dating app users and their degree of dating app use, they found that dating app user social anxiety and depressive symptoms were positively correlated with their level of dating app use, and that these symptoms predicted that men were less likely to initiate contact with people matched by dating apps, but not women.

In some research work, researchers have proposed methods or tools to help operators of apps, websites, hotel etc. to analyze user reviews. Considering that user reviews for apps are valuable for app operators to improve user experience and user satisfaction, but manually analyzing large numbers of user reviews to get useful opinions is inherently challenging, Vu et al. [24] proposed MARK, a keyword-based semi-automated review analysis framework that can help app operators analyze user reviews more effectively to get useful input from users. Jha and Mahmoud [25] proposed a novel semantic approach for app review classification, it can be used to extract user needs from application evaluations, enabling a more efficient classification

process and reducing the chance of overfitting. Dalal and Zaveri [26] proposed a view mining system for binary and fine-grained sentiment classification that can be used for user reviews, and empirical studies show that the proposed system can perform reliable sentiment classification at different granularity levels. Considering that a large number of user reviews need to be explored, analyzed, and organized to better assist website operators in making decisions, Sharma, Nigam and Jain [27] proposed an aspect-based opinion mining system to classify reviews, and empirically demonstrated the effectiveness of this system. Considering that hotel managers in Bali can gain insight into the perceived state of the hotel through hotel user reviews, Prameswari, Surjandari and Laoh [28] used text mining methods and aspect-based sentiment analysis in their research to capture hotel user opinions in the form of emotions. The results show that the Recursive Neural Tensor Network (RNTN) algorithm performs well in classifying the sentiment of words or aspects. As a result, we wish to applying machine learning models on mining user reviews of dating apps. In this way, operators of apps can better manage their user review data and improve their apps more effectively.

Considering the increasing popularity of dating apps and the unsatisfactory user reviews of major dating apps, we decided to analyze the user reviews of dating apps using two text mining methods. First, we established a topic model based on LDA to mine the negative reviews of mainstream dating apps, analyzed the main reasons why users give negative reviews, and put forward corresponding improvement suggestions. Next, we built a two-stage machine learning model that combined data dimensionality reduction and data classification, hoping to obtain a classification that can effectively classify user reviews of dating apps, so that app operators can process user reviews more effectively.

## 2 Data acquisition and research design

### 2.1 Data acquisition

At present, there are several dating apps that are widely used, such as the famous Tinder and Okcupid. Since most users download these apps from Google Play, we believed that app reviews on Google Play can effectively reflect user feelings and attitudes toward these apps. All the data we used are from reviews of users of these six dating apps: Bumble, Coffee Meets Bagel, Hinge, Okcupid, Plenty of Fish and Tinder. The data are published on figshare.com [29], we promise that sharing the dataset on Figshare complies with the terms and conditions of the sites from which data was accessed. Also, we promise that the methods of data collection used and its application in our study comply with the terms of the website from which the data originated. The data include the text of the reviews, the number of likes the reviews get, and the reviews' ratings of the apps. At the end of May 2022, we have collected a total of 1,270,951 reviews data. First of all, in order to prevent the impact on the results of text mining, we first carried out text cleaning, deleted symbols, irregular words and emoji expressions, etc.

Considering that there may be some reviews from bots, fake accounts or meaningless duplicates among the many reviews, we believed that these reviews can be filtered by the number of likes they get. If a review has no likes, or just a few likes, it can be considered that the content contained in the review is not of sufficient value in the study of user reviews, because it can't get enough commendations from other users. In order to keep the size of data we finally use not too small, and to ensure the authenticity of the reviews, we compared the two screening methods of retaining reviews with a number of likes greater than or equal to 5 and retaining reviews with a number of likes greater than or equal to 10. Among all the reviews, there are 25,305 reviews with 10 or more likes, and 42,071 reviews with 5 or more likes.

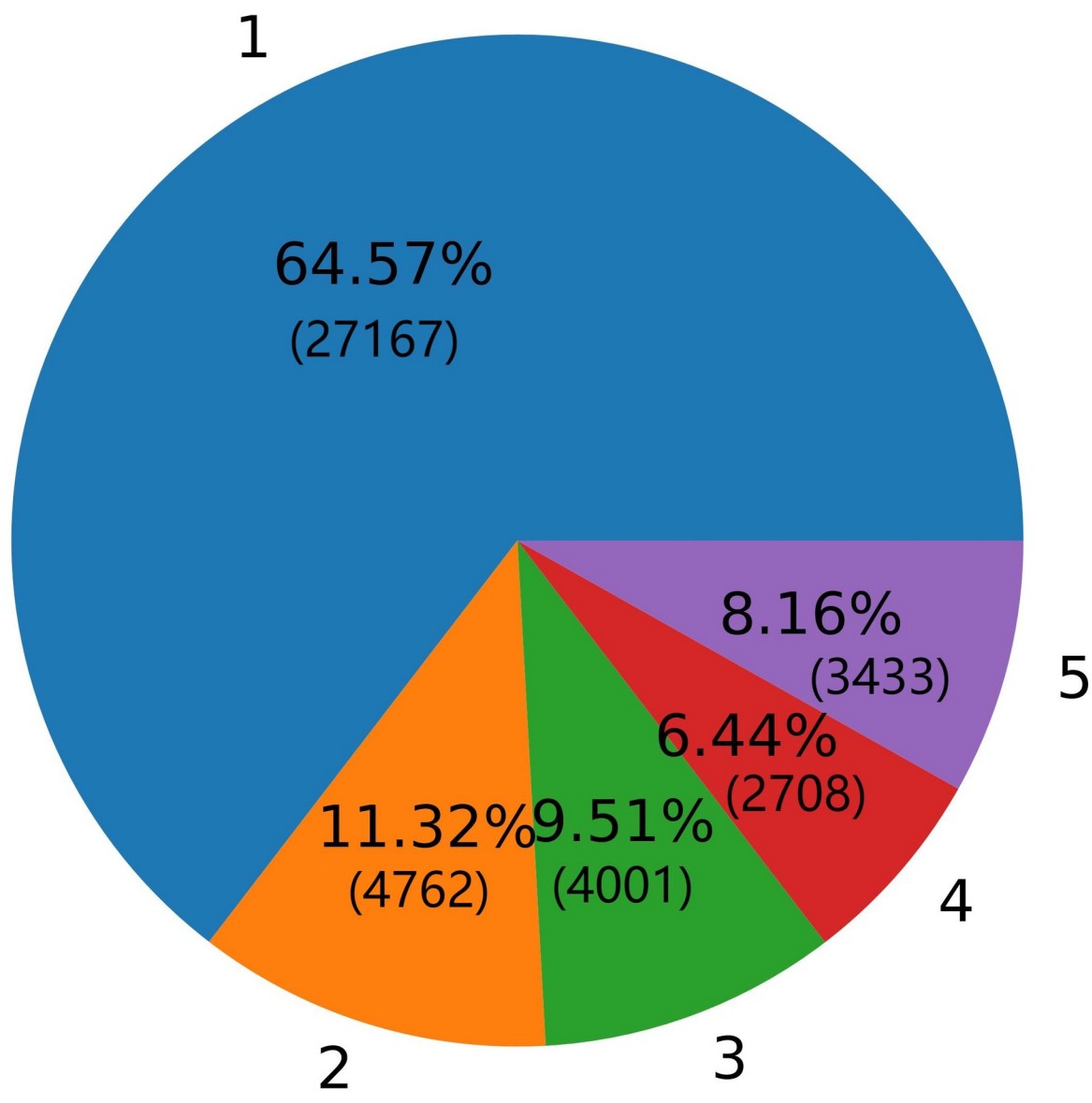

**Fig 1. Distribution of ratings for dating apps.**

In order to maintain a certain generality and generalizability of the results of the topic model and classification model, it is considered that relatively more data is a better choice. Therefore, we selected 42,071 reviews with a relatively large sample size with a number of likes greater than or equal to 5. In addition, in order to ensure that there are no worthless comments in the filtered comments, such as repeated negative comments from robots, we randomly selected 500 comments for careful reading and found no obvious worthless comments in these reviews. For these 42,071 reviews, we plotted a pie chart of reviewers' ratings of these apps, and the numbers such as 1,2 on the pie chart means 1 and 2 points for the app's ratings.

Looking at Fig 1, we find that the 1-point rating, which represents the worst review, accounts for the majority of the reviews on these apps; while all of the percentages of other

ratings are all less than 12% of the reviews. Such a ratio is very shocking. Most of the users who reviewed on Google Play were very dissatisfied with the dating apps they were using.

## 2.2 Research design

Considering the disparity between positive and negative reviews in dating app user reviews, we believed that we need to establish a topic model based on LDA, hoping to mine the customer opinions contained in the negative reviews. At the same time, we thought it is helpful to build a text emotion classification model that can better match dating app user reviews with ratings in Google Play on a scale of 1 to 5. Such a model can help the operators of dating apps to quickly and effectively classify the text of user reviews when conducting market research or collecting various user opinions, so as to mine customers' opinions which is reflected in positive, negative or moderate reviews with more details.

## 3 User negative review mining based on topic model

From the previous visualization results, we can clearly find that in the reviews of users on dating apps in Google Play, negative reviews account for the vast majority. This situation may have a certain impact on the operation of the apps, such as the loss of users or the continuous decline of the reputation of the apps. In order to solve such a problem, we have established a topic model using the implicit Dirichlet distribution to mine the information in the bad reviews of dating apps, in order to analyze the reasons behind the bad reviews and put forward corresponding suggestions that can solve the problem.

### 3.1 Topic model

With the explosive growth of text information, digging its theme composition from massive non-structured text information as the main mode of text information analysis. Topic Models are important type of machine learning algorithm that can discover potential mixed themes from the existing document set, so they are important unsupervised learning tools for inferring knowledge. Topic models are also universal ways to understand collections of unstructured text documents, they can be used to automate the screening of large amounts of text data. Once key topics have been identified, text files can be grouped for further analysis. There are many algorithms that can be applied to topic modeling, common ones are LSA, LDA (Latent Dirichlet Allocation), etc. In many research works that also use topic models such as [30–33], the effectiveness of LSA and LDA may be difficult to determine Consider that LDA considers both the distribution of topics by document and the distribution of terms by topic compared to LSA [34], we have applied Latent Dirichlet assignments.

The topic model LDA is a non-supervised machine learning method. It can effectively extract the hidden themes in large-scale document sets and corpus libraries. The dimensional reduction capacity, modeling ability and scalability have made it one of the popular research directions in the field of theme mining in recent years.

### 3.2 Latent Dirichlet Allocation

Dirichlet distribution is a probability distribution of a diverse continuous random variable, which is an extension of Beta Distribution. In Bayesian learning, Dirichlet distribution is often used as a prior distribution of multinomial distributions. Definition: The probability density

function of the multivariate continuous random variableis $\theta = (\theta_1, \theta_2, \ldots, \theta_n)$ is

$$p(\theta|\alpha) = \frac{\Gamma(\sum_{i=0}^{k} \alpha_i)}{\prod_{i=1}^{k} \Gamma(\alpha_i)} \prod_{i=1}^{k} \theta_i^{\alpha_i - 1} \tag{1}$$

$$\sum_{i=1}^{k} \theta_i = 1, \theta_i \geq 0, \alpha_i \geq 0,$$

Be written as

$$\theta(\alpha)$$

The gamma function is [35]

$$\Gamma(s) = \int_0^\infty x^{x-1} e^{-x} dx, s > 0 \tag{2}$$

The underlying Dirichlet assignment aims to represent each document T as a mixed distribution of topics, and each topic as a mixed distribution on dictionary D by modeling, where the input document set T and the number of topics S are given.

In addition to infering the topic distribution of words in a document, LDA makes assumptions about how topics and documents are generated. Let sigma represent the mixing ratio of words in S topics, a matrix of size S by D. The s-row of the matrix corresponds to the distribution $\sigma_s(\alpha)$ of words on topic S, which means that each $\sigma_i$ is extracted from a Dirichlet distribution with a symmetric hyperparameter $\alpha$. $\phi$ is the topic mixing ratio of the document and is a $T * S$ matrix. Where $\phi_t$ in line T corresponds to the topic mixing ratio of document T, and $\phi_t(\beta)$ means that each $\phi_t$ is extracted from the Dirichlet distribution with symmetric hyperparameter $\beta$. When generating the n word in the document t, extract a topic according to the topic distribution $\phi_t$ of this document as the corresponding topic of this word, numbered $z_t n$ ($\phi_t$) at first. Then, according to the word distribution $\omega_t n$ of this topic, the word $\omega_t n$ is extracted and numbered $\omega_t n(\sigma_{zm})$. A complete document can be generated by repeated multiple times, while the entire corpus can be generated by repeated extraction of documents [36].

## 3.3 Determination of hyperparameters

In order for the model to obtain better analysis results, we first need to determine the values of the hyperparameters of the topic model based on LDA. $\alpha$ and $\beta$ are main hyperparameters, and we first considered the choice of hyperparameter $\beta$. As a common setting in Griffiths and Steyvers [37], we set $\beta = 0.01$. To the hyperparameter $\alpha$, we also have considered the method to calculate this hyperparameter from Griffiths and Steyvers [37] that $\alpha = 50/t$ (where t is number of topics) as the baseline method. Since we consider 27,167 reviews with a score of 1 for analysis, the sample size is not very large, we believed the number of topics shouldn't be greater than 50.

Topic coherence based on word co-occurrence patterns is one of the effective indicators to measure the effect of topic models. Therefore, we set different hyperparameters based on [37], obtained the topic coherence scores corresponding to these hyperparameters, and presented the results in the form of a line chart in Figs 2 and 3.

Figs 2 and 3 shows that the baseline coherence scores of the hyperparameters calculated according to the method proposed by Griffiths and Steyvers [37] are around 0.2 with all the choices of topics' numbers. Since we don't need to many topics, we considered smaller $\alpha$. And the topic coherence scores corresponding to smaller $\alpha$ are shown in Figs 4 and 5.

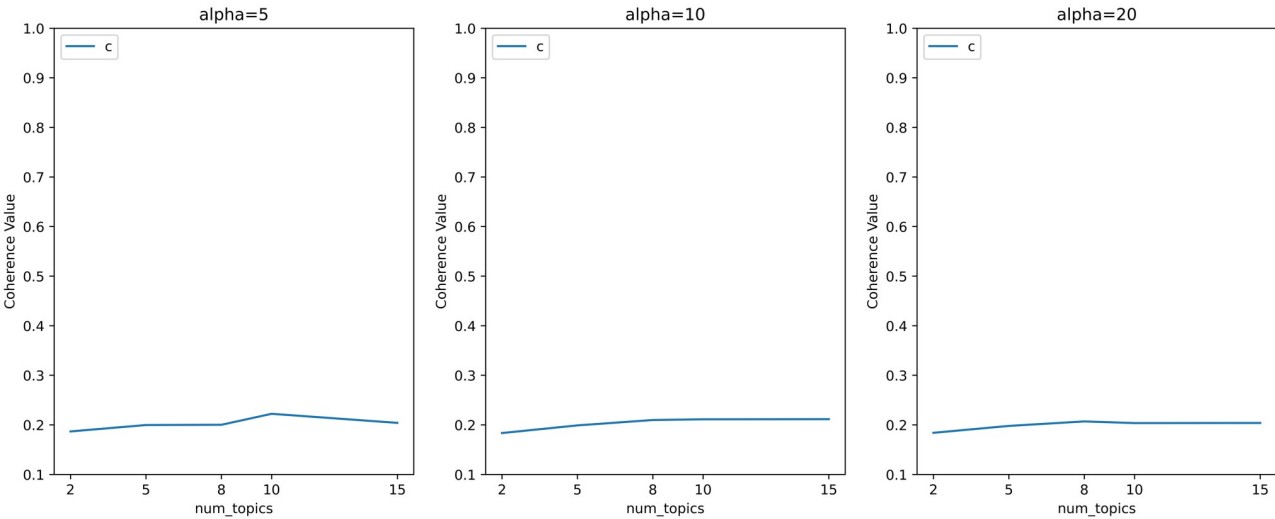

**Fig 2. Line chart of hyperparametric search for topic models with less topics and high $\alpha$.**

According to the results showed in Figs 4 and 5, we believed that when the hyperparameter $\alpha = 0.01$ and the number of topics is 10, the model can obtain the highest topic coherence score among the four figures. The final calculation results in topic coherence score = 0.469 is around twice the value of the baseline coherence scores. So we finally chose the LDA topic model with $\alpha = 0.01$, $\beta = 0.01$ and 10 topics.

## 3.4 Analysis of the topic modeling results

Table 1 shows the analysis results of the LDA topic model with the final hyperparameters.

- Topic 1 shows the words 'swipe' 'pay' 'match' 'gold' 'money' 'useless' 'buy'. There are complaints from users about the matching system and matters in payment.

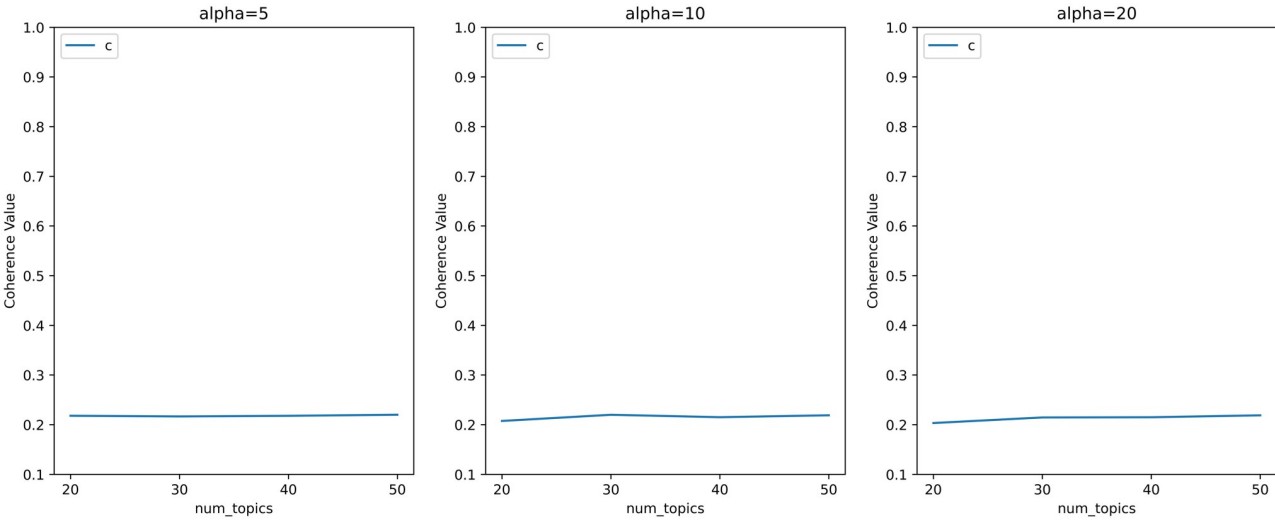

**Fig 3. Line chart of hyperparametric search for topic models with more topics and high $\alpha$.**

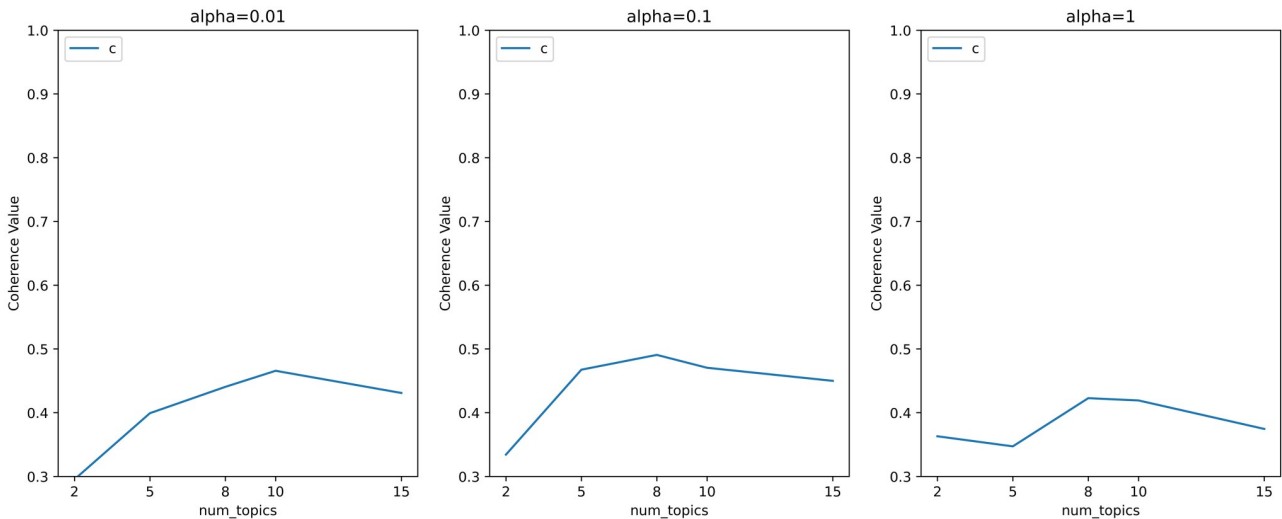

**Fig 4. Line chart of hyperparametric search for topic models with less topics and low $\alpha$.**

- Topic 2 shows the words 'match' 'open' 'say' 'notification' 'download' 'contact'. This topic doesn't focus on a certain problem.

- Topic 3 shows the words 'wrong' 'woman' 'mile' 'date' 'distance' 'boost'. We believe these words indicate user dissatisfaction with long distances matching and women's rights in dating.

- Topic 4 shows the words 'super' 'old' 'ridiculous' 'ad' 'block'. These words reflect that users have great opinions on the advertising push mechanism of apps.

- Topic 5 shows the words 'login' 'error' 'code' 'photo' 'scam'. Users complained about errors when logging in and false information from other users and the resulting scams.

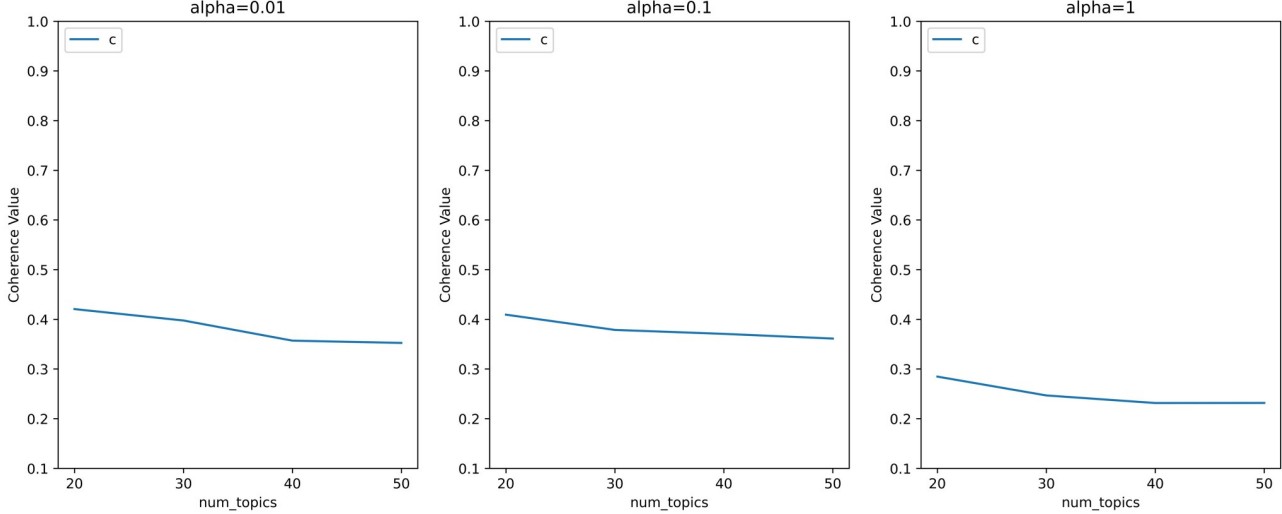

**Fig 5. Line chart of hyperparametric search for topic models with more topics and low $\alpha$.**

**Table 1. Analysis results of the LDA topic model.**

| Topic No. | Keywords | Topic Summary |
|---|---|---|
| 1 | swipe, pay, match, gold money, useless, buy, ⋯ | Matching system and payment |
| 2 | match, open, notification say, download, contact, ⋯ | No notable problems |
| 3 | wrong, woman, mile, date distance, boost, ⋯ | Long distance match women's rights |
| 4 | super, old, ridiculous ad, block, ⋯ | Ad push |
| 5 | login, error, code, photo, scam, ⋯ | Login problem and scams from fake accounts |
| 6 | ban, spam, stupid, rate bot, write, report, ⋯ | More effective communication |
| 7 | fake, profile, account, reply cancel terrible, ⋯ | User authenticity and account issues |
| 8 | completely, system, help end, everytime, ⋯ | Hope the apps are more functional |
| 9 | uninstalled, membership, install, screen, joke, face, ⋯ | Issues with fake accounts and apps installation |
| 10 | pay, subscription, money, bad, charge, refund, purchase, ⋯ | Dissatisfaction of payment and refund |

- Topic 6 shows the words 'spam' 'rate' 'write' 'report'. From these words, We believe that users expect more effective communication.

- Topic 7 contains "fake' 'profile' 'account' 'reply' 'cancel' 'terrible'. Users want the authenticity of users on apps to be guaranteed and are concerned about issues with their accounts.

- Topic 8 with words such as 'completely' 'system' 'everytime' "end' 'help' reveals that users think the existing functions of dating apps are not complete, and they want dating apps to be more systematic and meet their needs at any time.

- Topic 9 contains "uninstalled' 'membership' 'install' 'screen' 'joke' 'face'. Users have complained about fake accounts in dating apps and problems with app downloads and uninstalls.

- Topic 10 with words like 'pay' 'subscription' 'money' 'charge' 'refund' 'purchase' 'bad'. It mainly reflects the user's dissatisfaction with matters related to payment and refund.

In general, the negative reviews of users are mainly concentrated in four aspects. First, they are dissatisfied with the payment amounts in dating apps and problems with the payment and refund processes; secondly, they complained about fake accounts in apps, including fake photos, personal information, and the resulting scams. This question is raised when people use many dating apps, Duguay [38] has pointed out that this problem exists in Tinder and has caused a certain negative impact; thirdly, they were dissatisfied with the advertising push mechanism and subscription mechanism in apps; finally, they also felt that the matching mechanism provided by apps needed to be improved.

## 4 Customer review classification based on two-stage machine learning

In order to establish a text sentiment classification model that can distinguish different reviews in a timely and effective manner, we first cleaned the comment text and perform term frequency–inverse document frequency (TF-IDF) processing on the data. The processed data are divided into training dataset and test dataset according to the ratio of 80% and 20%. Looking at the dimensionalities of the training data set, we find that the training dataset is higher than 25,000 dimensionalities, which is close to the number of samples in the training data set itself.

We believed that the high dimensionality of the training dataset would interfere with the accuracy of the classification model. At the same time, as shown in Fig 1, the proportion of each category in the dataset is very uneven, which may also interfere with the accuracy of the classification model. In addition, we considered the methods of applying XGBoost and LightGBM to learn only oversampled data as baseline models to evaluate the effect of applying dimensionality reduction models and the final effect of two-stage models.

## 4.1 Data dimensionality reduction and oversampling

**4.1.1 Principal component analysis.** Principal Component Analysis (PCA) is a famous data dimensionality reduction model in multivariate statistical analysis. It extracts the amount of information contained in the original high-dimensional data by linear combination, which is called principal components. According to the variance contribution rate of the extracted principal components, the principal component analysis selects the amount of information required in practical applications to achieve the purpose of reducing the data dimension.

①we normalize the data. For the matrix formed by the word vector:

$$X = \begin{bmatrix} x_{11} & x_{12} & \cdots & x_{1p} \\ x_{21} & x_{22} & \cdots & x_{2p} \\ \vdots & \vdots & \ddots & \vdots \\ x_{n1} & x_{n2} & \cdots & x_{np} \end{bmatrix}$$

Then we can standardize the data according to formula 3:

$$\mathrm{x}_{ij}^* = \frac{x_{ij} - \hat{x}_j}{\sqrt{Var(x_j)}} \tag{3}$$

With $\hat{x}_j = \frac{1}{n} \sum_{i=1}^{n} x_{ij}$, $Var(x_j) = \frac{1}{n-1} \sum_{i=1}^{n} (x_{ij} - \hat{x}_j)^2$, $(j = 1, 2, \cdots, p)$

②Let $X^*$ represent the matrix after data normalization, then the expression of the correlation coefficient matrix $R$ between these data is:

$$R = \begin{bmatrix} r_{11} & r_{12} & \cdots & r_{1p} \\ r_{21} & r_{22} & \cdots & r_{2p} \\ \vdots & \vdots & \ddots & \vdots \\ r_{p1} & r_{p2} & \cdots & r_{pp} \end{bmatrix}$$

With $r_{ij} = \frac{Cov(x_i^*, x_j^*)}{\sqrt{Var(x_1^*)}\sqrt{Var(x_2^*)}}$, $(n > 1)$.

③Calculate the eigenvalues$(\lambda_1, \lambda_2, \cdots, \lambda_p)$ and eigenvectors of the correlation coefficient matrix R:

$$\mathrm{a}_i = (a_{i1}, a_{i2}, \cdots, a_{ip})(i = 1, 2, \cdots, p) \tag{4}$$

④Next, we should select the required principal components and derive the expression for the principal components. The variance of each principal component obtained by the principal component analysis and the information contained in the principal component are decreasing

from large to small. We follow the variance contribution rate of each principal component:

$$variance \quad contribution rate = \frac{\lambda_i}{\sum\limits_{i=1}^{p} \lambda_i} \tag{5}$$

Here we select the principal components with variance contribution rate in the top 90% to build a new machine learning classification model.

⑤Finally, calculate the specific values of the principal components we need. Substitute the normalized data for calculating the principal components:

$$F_{ij} = a_{j1}x_{i1}^* + a_{j2}x_{i2}^* + \cdots + a_{jp}x_{ip}^* \tag{6}$$

$$(i = 1, 2, \cdots, n; j = 1, 2, \cdots, k)$$

Finally, we get the required dimensionality-reduced data, which is expressed in matrix form as [39]:

$$\begin{bmatrix} F_{11} & F_{12} & \cdots & F_{1k} \\ F_{21} & F_{22} & \cdots & F_{2k} \\ \vdots & \vdots & \ddots & \vdots \\ F_{p1} & F_{p2} & \cdots & F_{pp} \end{bmatrix}$$

**4.1.2 Locally linear embedding.** Locally Linear Embedding (LLE) is a method of data dimensionality reduction for the data of nonlinear signal feature vector dimensions. The idea of this dimensionality reduction is not just to reduce the number of dimensions, but to maintain the mathematical properties of the original data. Under the condition that the characteristics remain unchanged, LLE maps the data in the high-dimensional space to the low-dimensional space, and realizes the second extraction of eigenvalues in the data.

For each training instance $x^{(i)}$, LLE analyzes the k nearest neighbors of the instance, and then tries to reconstruct $x^{(i)}$ as a linear combination of these neighbors. More specifically, LLE makes the Euclidean distance between $x^{(i)}$ and $\sum\limits_{j=1}^{m} w_{i,j}x^{(j)}$ as small as possible by finding the weight $w_{i,j}$. Meanwhile LLE assumes $w_{i,j} = 0$ if $x^{(j)}$ is not one of the k nearest neighbors of $x^{(i)}$. Therefore, LLE first constructs an unconstrained optimization, where W is a weight matrix containing all weights $w_{i,j}$, and the second constraint requires that the weights of each training data $x^{(i)}$ be normalized.

$$\hat{W} = argmin \sum\limits_{i=1}^{m} (x^{(i)} - \sum\limits_{j=1}^{m} w_{i,j}x^{(j)})^2 \tag{7}$$

$$s.t. \begin{cases} w_{i,j} = 0, x^{(j)} \notin c.n. \quad of \; x^{(i)} \\ \sum\limits_{j=1}^{m} w_{i,j} = 1, i = 1, 2, \cdots, m \end{cases} \tag{8}$$

Then, the weight matrix $\hat{W}$ (containing the weights $\hat{w}_{i,j}$) will encode the local linear relationship between the training data.

Next, LLE will map the training data to a d-dimensional space (where *dn*), while preserving these local relationships as much as possible. Considering that $z^{(i)}$ is the dimensionality reduction result of $x^{(i)}$ in this d-dimensional space, we hope that the Euclidean distance between $z^{(i)}$ and $\sum_{j=1}^{m} \hat{W}_{i,j} z^{(j)}$ is as small as possible, thereby updating the previous objective function. LLE doesn't take the approach of finding the best weights while keeping the data fixed here. In contrast, LLE keeps the weights constant and finds the best location for the data in a low-dimensional space. Here Z is a matrix containing all $z^{(i)}$. [40]

$$\hat{Z} = argmin \sum_{i=1}^{m} (z^{(i)} - \sum_{j=1}^{m} \hat{W}_{i,j} x^{(j)})^2 \tag{9}$$

**4.1.3 Oversampling.** Before building a classifier to fit the dimensionality-reduced data, we should pay attention to the serious imbalance of classification labels in the data. It can be intuitively found from the previous pie chart Fig 1 that the number of samples with classification labels 2, 3 and 4 is much less than the number of samples with labels 1 and 5. In general, such a situation can easily affect the performance of the classification model. The research of Chawla et al. [41] proposed the SMOTE oversampling method and confirmed that this method can avoid overfitting caused by traditional oversampling. In order to solve this problem, we have used the SMOTE method to oversample the data when building the classification model, so that the sample size corresponding to each classification label tended to be balanced.

## 4.2 Classification models and results

**4.2.1 Model introduction.** Both XGBoost and LightGBM are powerful ensemble learning models further improved on the basis of Gradient Boosting Decision Tree (GBDT). XGBoost was proposed by Chen and Guestrin [42]. Compared with GBDT, XGBoost adds a regularization part to the loss function to achieve better generalization ability. At the same time, the loss function of XGBoost performs second-order Taylor expansion on the error part, which is more accurate than the first-order Taylor expansion on the error part in GBDT. XGBoost performs parallel selection on the establishment process of each weak learner in the algorithm, which improves the efficiency of the algorithm. Afifah [43] performed sentiment analysis on reviews of a telemedicine application in Indonesia using XGBoost and achieved over 96% classification accuracy. In Zhang's research [44], XGBoost can perform the task of criminal text classification more effectively than other models.

LightGBM was proposed by Ke et al. [45]. Its internal decision trees use a leaf-wise growth strategy and sets the *max_depth* hyperparameter to prevent overfitting. LightGBM also uses the histogram algorithm. According to the data binning strategy, the nodes of the decision tree can improve the calculation speed when splitting. The two learning methods it supports feature parallelism and data parallelism further reduce the computational cost. In addition, the Gradient-based One-Side Sampling and Exclusive Feature Bundling algorithms included in LightGBM have both contributed to improving the classification accuracy. He et al. [46] believed that the LightGBM works well on social media prediction tasks. And the research of Zvonarev and Bilyi [47] showed that LightGBM has good performance in sentiment analysis of Russian texts.

**4.2.2 Model evaluation.** The results of our model are presented in Table 2. Data dimensionality reduction and oversampling using principal component analysis did not improve the classification effect of the LightGBM model much. But for the XGBoost model

**Table 2. Model accuracy before and after oversampling.**

|  | XGBoost | LightGBM |
|---|---|---|
| Original | 0.697 | 0.688 |
| PCA and oversampling | 0.883 | 0.725 |
| LLE and oversampling | 0.718 | 0.701 |

with its hyperparameters max depth = 10, n estimators = 50 and learning rate = 0.1, such data processing improved the accuracy of the classification model to 0.883. After data dimensionality reduction of PCA and oversampling, the areas under the ROC curve of the LightGBM and XGBoost models at Figs 6 and 7 perform better on each class. In particular, the area under the ROC curve of the XGBoost model exceeds 0.9 for each class. The confusion matrices in Figs 8 and 9 are also strong evidence that the XGBoost model achieves good results.

On the other hand, 0.883 is close to 0.19 more accurate than using XGBoost's baseline model. Moreover, as the data has a total of 5 categories was considered, we believed that this level of accuracy is high enough to be acceptable. We suggest that dating app operators can use this method to establish an XGBoost model on the data after dimensionality reduction by principal component analysis and oversampling, so as to facilitate batch text sentiment classification for customer opinions collected in the future.

## 5 Conclusion and outlook

Regarding the problems analyzed from the user's negative reviews on the current mainstream dating apps, we make the following suggestions for the operators of the apps:

At first, we suggest that the operators of apps can conduct a questionnaire survey on users to understand the user's expectations for the payment amount, and may try to charge

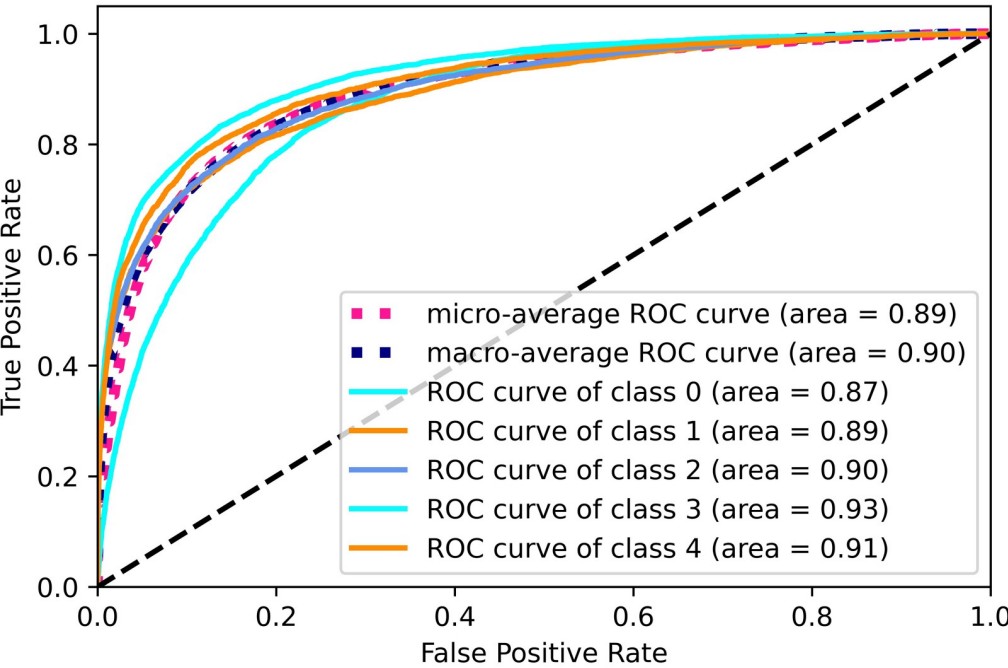

**Fig 6. ROC curve for PCA and LightGBM.**

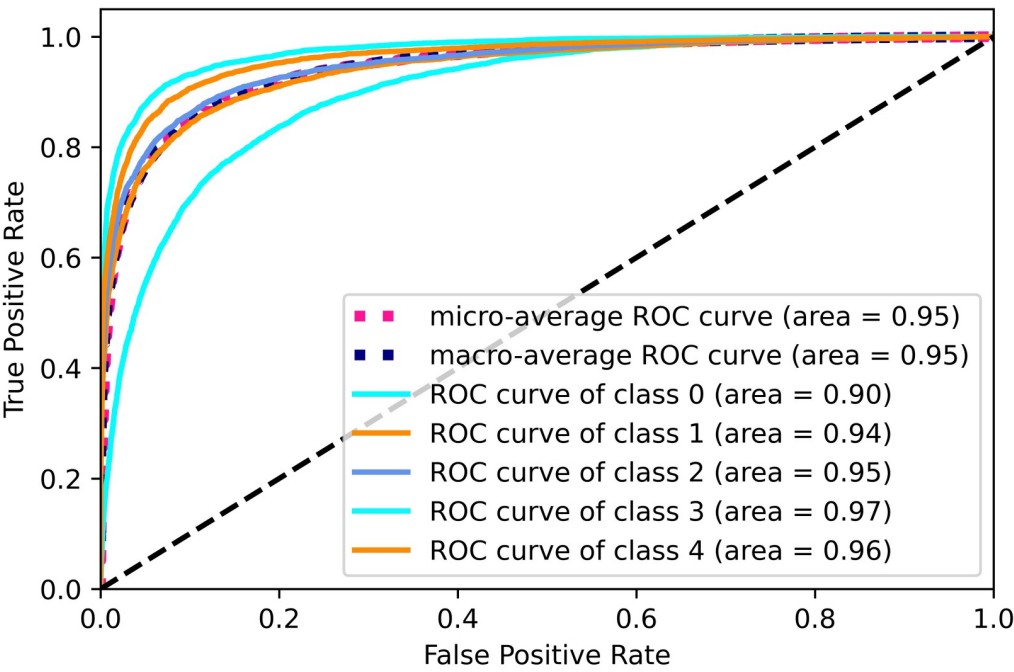

**Fig 7. ROC curve for PCA and XGBoost.**

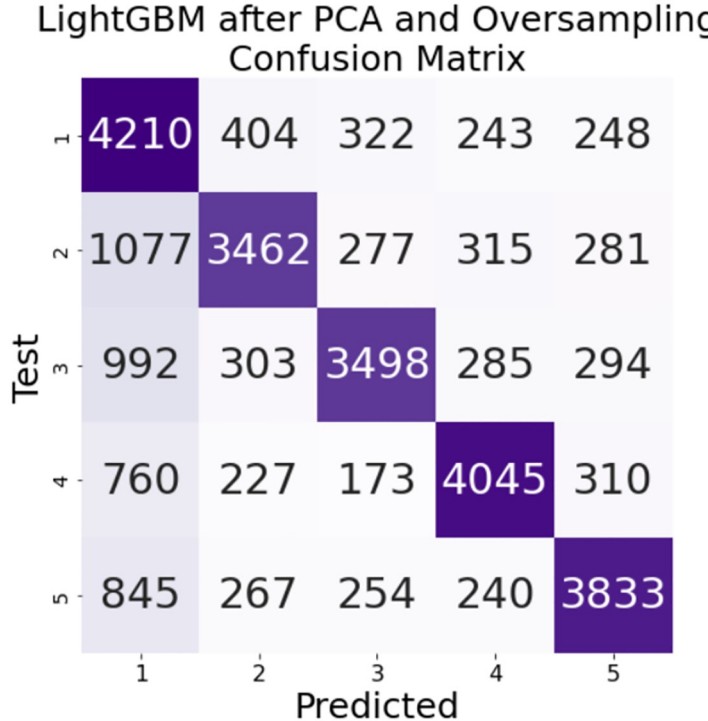

**Fig 8. Confusion matrix curve for PCA and LightGBM.**

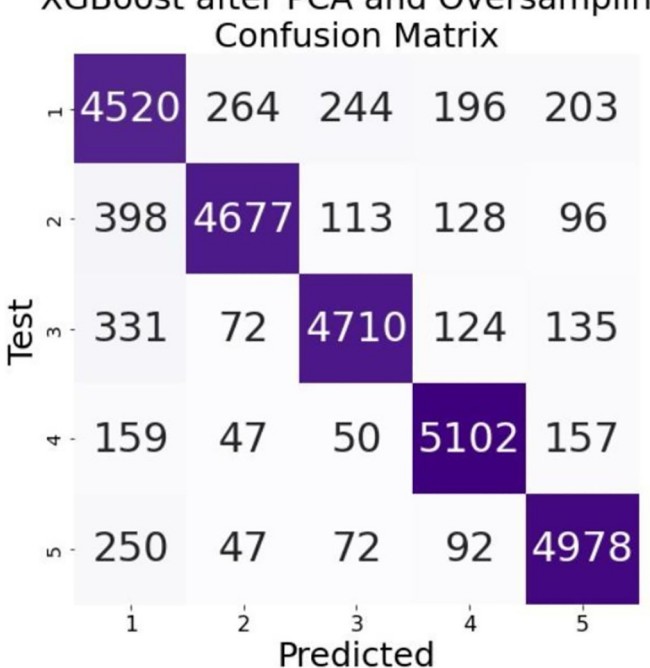

**Fig 9. Confusion matrix for PCA and XGBoost.**

according to the user's needs. At the same time, the current payment and refund system should be improved to ensure the timely completion of payment and refund for users.

Secondly, we recommend that operators of apps develop more effective ways to identify false information about users. For example, the authenticity of user information can be analyzed by combining multi-modal data such as user photos, IP addresses, and login times.

Then, we suggest that app operators improve the current advertising push mechanism and subscription mechanism, and they can develop more effective recommendation system algorithms to meet different user needs.

Futhermore, we recommend that the operators of apps improve the dating object matching mechanism, which can start from the distance between users, the ages and living habits of users, etc., and also increase the dominance of female users in matching.

Finally, we believe that it is more effective to use principal component analysis to reduce the dimensionality of the text vectors of user reviews of dating apps, and then use the XGBoost model to learn the oversampled low-dimensional data. Such a two-stage machine learning model can efficiently classify these reviews so that operators of apps can batch process the collected user review data.

## Supporting information

**S1 Graphical abstract.**
(TIF)

## Author Contributions

**Conceptualization:** Qian Shen, Siteng Han.

**Data curation:** Qian Shen, Siteng Han, Yu Han.

**Formal analysis:** Qian Shen, Siteng Han, Yu Han, Xi Chen.

**Funding acquisition:** Xi Chen.

**Investigation:** Qian Shen, Siteng Han.

**Methodology:** Qian Shen, Siteng Han, Xi Chen.

**Project administration:** Xi Chen.

**Resources:** Xi Chen.

**Software:** Qian Shen.

**Supervision:** Qian Shen, Xi Chen.

**Validation:** Qian Shen, Yu Han.

**Visualization:** Qian Shen, Siteng Han, Yu Han.

**Writing – original draft:** Qian Shen, Siteng Han, Yu Han.

**Writing – review & editing:** Qian Shen, Xi Chen.

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
