## [Decision Letter · Decision Letter 0]

4 Dec 2022

PONE-D-22-30174Analysis of Dating Apps' User Reviews Based on Text MiningPLOS ONE

Dear Dr. Chen,

Thank you for submitting your manuscript to PLOS ONE. After careful consideration, we feel that it has merit but does not fully meet PLOS ONE’s publication criteria as it currently stands. Therefore, we invite you to submit a revised version of the manuscript that addresses the points raised during the review process.

We look forward to receiving your revised manuscript.

Kind regards,

Viacheslav Kovtun, Dr.Sc., Ph.D.

Academic Editor

PLOS ONE

Journal Requirements:

2. In your Methods section, please include additional information about your dataset and ensure that you have included a statement specifying whether the collection method of the collected dataset, and its use for research, complied with the terms and conditions for the website.

3. Please note that PLOS ONE has specific guidelines on code sharing for submissions in which author-generated code and data that underpins the findings in the manuscript. In these cases, all author-generated code must be made available without restrictions upon publication of the work. Please review our guidelines at https://journals.plos.org/plosone/s/materials-and-software-sharing#loc-sharing-code and ensure that your code is shared in a way that follows best practice and facilitates reproducibility and reuse. New software must comply with the Open Source Definition. If data cannot be openly shared for ethical or legal restrictions, please provide details for accessing these data from the original data holder.

"The work was financialy supported by Chinese National Natural Science Foundation with Grant No.12101473, the Basic Research Program of Natural Science in Shaanxi Province with Grant Nos.2021JQ-764 and 2021JQ-766, and Education Department of Shaanxi Province with Grant No.21JK0640."

"Xi Chen was financialy supported by Chinese National Natural Science Foundation with Grant No.12101473, the Basic Research Program of Natural Science in Shaanxi Province with Grant Nos.2021JQ-764 and 2021JQ-766, and Education Department of Shaanxi Province with Grant No.21JK0640.

7. We note you have included a table to which you do not refer in the text of your manuscript. Please ensure that you refer to Table 1 in your text; if accepted, production will need this reference to link the reader to the Table. 

Reviewers' comments:

Reviewer's Responses to Questions

**Comments to the Author**

1. Is the manuscript technically sound, and do the data support the conclusions?

Reviewer #1: No

Reviewer #2: Yes

2. Has the statistical analysis been performed appropriately and rigorously? 

Reviewer #1: No

Reviewer #2: Yes

3. Have the authors made all data underlying the findings in their manuscript fully available?

Reviewer #1: Yes

Reviewer #2: Yes

4. Is the manuscript presented in an intelligible fashion and written in standard English?

Reviewer #1: No

Reviewer #2: Yes

5. Review Comments to the Author

Reviewer #1: It's possible that I'll suggest giving this article a presentation at a conference so that You may have an audience and continue the conversation about the issues. My primary justification is on the fact that "users" are the "unit of analysis" linked with this study; hence, there needs to be a concrete subjective evaluation to generate an accurate construct that is suitable for quantifying the impact of "User Reviews." However, while employing machine learning, the assumption is that a new course of action "or knowledge" can emerge from the analysis of the machine learning, sadly the work has not showed any improvement on the problems linked with "User Reviews" on dating apps. Generally speaking, user-based aspects used in this study is faulty.

You don't need any sort of conceptualised form of an advanced thought in order to gain something new through "Negative Review Mining." Instead, all you need is to have an open mind.

To begin, you have to get a firm grasp on the fact that ratings and reviews are inextricably linked, and it is your responsibility to ensure that this connection is maintained.

The users review and rating is important because it has led to understanding of the rating approached mapped to reviews.

Rating is numeric and reviews are reflections. The numeric values of rating will indicate if DATIN APPS is Good or BAD, whereas reviews will indicate the user’s perceptions. Previous research reveals that “Bad ratings are trustworthy regardless of the number of reviews”, that is users tend to believe reported bad rating. On the other hand, “Good ratings are trustworthy only when they come along with a high number of reviews”

The proposed four facets of "bad reviews of users" are not utilised in any novel way by Section 3, which does not produce anything that is formulated.

The fourth section is a routine analysis, and it does not introduce anything novel that contributes to the advancement of the research field. The XGBoost and LightGBM models are applied to the dataset that is already in existence without any additional information being connected to them.

,

Reviewer #2: In this paper, the authors are proposed “Analysis of Dating Apps’ User Reviews Based on Text Mining”.

The strengths of the paper are that it is well structured, the description of the related work is well done and that results are extensively compared to results of the similar research.

Minor revisions:

1. Authors should draw a graphical abstract of the proposed approach

2. Authors should mention the names of the all the six dating apps

3. Why authors used LDA instead of LSA.

4. Why authors used XGBoost justify it

5. Proofread the entire manuscript

6. Authors follow the recently published paper “Abdulkadhar, S., Murugesan, G., & Natarajan, J. (2020). Classifying protein-protein interaction articles from biomedical literature using many relevant features and context-free grammar. Journal of King Saud University-Computer and Information Sciences, 32(5), 553-560.”

7. Proofread the entire manuscript.

6. PLOS authors have the option to publish the peer review history of their article (what does this mean?). If published, this will include your full peer review and any attached files.

Reviewer #1: No

Reviewer #2: No

---

## [Author Response · Author response to Decision Letter 0]

14 Feb 2023

#Responses to Academic Editor:

Thank you very much for your guidance and comments on our work. Please find our itemized responses below and our corrections in the re-submitted files.

1. Comments: Please ensure that your manuscript meets PLOS ONE's style requirements, including those for file naming. The PLOS ONE style templates can be found at 

Response: We used the latex template provided by PLOS ONE for our writing and followed the format of the paper required by PLOS ONE.

2. Comments: In your Methods section, please include additional information about your dataset and ensure that you have included a statement specifying whether the collection method of the collected dataset, and its use for research, complied with the terms and conditions for the website.

Response: We have uploaded the dataset to figshare.com and made it publicly available, referenced the dataset in the Data Acquisition section of the article, and added a statement of compliance with data collection. The URI of our data is:

https://figshare.com/articles/dataset/Text_of_user_reviews_of_dating_apps/21895827

3. Comments: Please note that PLOS ONE has specific guidelines on code sharing for submissions in which author-generated code and data that underpins the findings in the manuscript. In these cases, all author-generated code must be made available without restrictions upon publication of the work. Please review our guidelines at https://journals.plos.org/plosone/s/materials-and-software-sharing#loc-sharing-code and ensure that your code is shared in a way that follows best practice and facilitates reproducibility and reuse. New software must comply with the Open Source Definition. If data cannot be openly shared for ethical or legal restrictions, please provide details for accessing these data from the original data holder.

Response: The code we used has been publicly posted on GitHub, and the URL of the code (https://github.com/Qian0214Shen/code-of-text-mining-paper) has been filled in the relevant field when submitted.

4. Comments: Thank you for stating the following in the Acknowledgments Section of your manuscript: 

"The work was financialy supported by Chinese National Natural Science Foundation with Grant No.12101473, the Basic Research Program of Natural Science in Shaanxi Province with Grant Nos.2021JQ-764 and 2021JQ-766, and Education Department of Shaanxi Province with Grant No.21JK0640."

"Xi Chen was financialy supported by Chinese National Natural Science Foundation with Grant No.12101473, the Basic Research Program of Natural Science in Shaanxi Province with Grant Nos.2021JQ-764 and 2021JQ-766, and Education Department of Shaanxi Province with Grant No.21JK0640.

Please include your amended statements within your cover letter; we will change the online submission form on your behalf..

Response: We have removed the section on funding information from the paper.

5. Comments: In your Data Availability statement, you have not specified where the minimal data set underlying the results described in your manuscript can be found. PLOS defines a study's minimal data set as the underlying data used to reach the conclusions drawn in the manuscript and any additional data required to replicate the reported study findings in their entirety. All PLOS journals require that the minimal data set be made fully available. For more information about our data policy, please see http://journals.plos.org/plosone/s/data-availability.

Response: We have uploaded the dataset to figshare.com and made it publicly available, referenced the dataset in the Data Acquisition section of the article, and added a statement of compliance with data collection. The URI of our data is:

https://figshare.com/articles/dataset/Text_of_user_reviews_of_dating_apps/21895827

6. Comments: PLOS requires an ORCID iD for the corresponding author in Editorial Manager on papers submitted after December 6th, 2016. Please ensure that you have an ORCID iD and that it is validated in Editorial Manager. To do this, go to ‘Update my Information’ (in the upper left-hand corner of the main menu), and click on the Fetch/Validate link next to the ORCID field. This will take you to the ORCID site and allow you to create a new iD or authenticate a pre-existing iD in Editorial Manager. Please see the following video for instructions on linking an ORCID iD to your Editorial Manager account: https://www.youtube.com/watch?v=_xcclfuvtxQ

Response: We have added ORCID information for the corresponding author in her account.

7. Comments: We note you have included a table to which you do not refer in the text of your manuscript. Please ensure that you refer to Table 1 in your text; if accepted, production will need this reference to link the reader to the Table

Response: We have already referenced Table 1 below it.

#Responses to Reviewer 1:

Thank you very much for taking your time to review this manuscript. Please find our itemized responses below.

1. Comments:  It's possible that I'll suggest giving this article a presentation at a conference so that You may have an audience and continue the conversation about the issues. My primary justification is on the fact that "users" are the "unit of analysis" linked with this study; hence, there needs to be a concrete subjective evaluation to generate an accurate construct that is suitable for quantifying the impact of "User Reviews." However, while employing machine learning, the assumption is that a new course of action "or knowledge" can emerge from the analysis of the machine learning, sadly the work has not showed any improvement on the problems linked with "User Reviews" on dating apps. Generally speaking, user-based aspects used in this study is faulty.

Response: The starting point of our research is based on the perspective of enterprise information management, that is, by mining the user reviews of the apps, analyzing how the operators of the apps can improve the apps based on the opinions of users, and trying to develop a method for the operators of the apps to quickly classify the unmarked user reviews collected. Regarding the impact of quantifying user reviews that you mentioned, this may require a large-scale market research to understand how a large number of users perceive these reviews in Google Play and how these reviews influence user choices. In fact, this is indeed a very interesting and exciting topic, and we hope to have the opportunity to collect more suitable data for such research in further research.

2. Comments: You don't need any sort of conceptualised form of an advanced thought in order to gain something new through "Negative Review Mining." Instead, all you need is to have an open mind.

Response: We strongly agree that an open mind is indeed a necessary condition for people to accept bad reviews, but in the current era of big data, it is difficult for app operators to rely only on an open mind to mine information from a large number of reviews. Therefore, we hope to efficiently mine information from massive user reviews by applying machine learning models.

3. Comments: Rating is numeric and reviews are reflections. The numeric values of rating will indicate if DATIN APPS is Good or BAD, whereas reviews will indicate the user’s perceptions. Previous research reveals that “Bad ratings are trustworthy regardless of the number of reviews”, that is users tend to believe reported bad rating. On the other hand, “Good ratings are trustworthy only when they come along with a high number of reviews”

Response: In the current reviews of apps, malicious low-scoring reviews or worthless reviews from bots are always inevitable, and these reviews are difficult to express the general thoughts of app users. There are other comments, perhaps due to their short publication time, that do not receive enough likes, and the value of these comments is difficult to determine in batches. Therefore, in order to control the mining value of the data and ensure that the size of data in the dataset is not too small to affect the fit of the machine learning model, we select comments with more than or equal to 5 likes for analysis.

4. Comments: The fourth section is a routine analysis, and it does not introduce anything novel that contributes to the advancement of the research field. The XGBoost and LightGBM models are applied to the dataset that is already in existence without any additional information being connected to them.

Response: We should admit that the fourth part of our study is only an application to existing methods. First of all, LightGBM and XGBoost are actually very good machine learning classification models and are widely used in machine learning research in various fields. And in the process of application, we obtained 88.3% good accuracy for machine learning classification tasks with 5 classes, which is close to 19% higher than our baseline model, so we think this result is acceptable.

#Responses to Reviewer 2:

Thank you very much for your guidance and comments on our work. Please find our itemized responses below and our corrections in the re-submitted files.

1. Comments: Authors should draw a graphical abstract of the proposed approach

Response: We have produced the Graphic Abstract and submitted it as an attachment.

2. Comments: Authors should mention the names of the all the six dating apps

Response: We have declared the names of 6 apps in the section of data acquisition in Page 4.

3. Comments: Why authors used LDA instead of LSA

Response: Comparing some previous studies, we think it is difficult to say which is better than LSA or LDA. But considering that LDA considers both the distribution of topics by document and the distribution of topics over LSA, we chose LDA, as detailed at the end of the first paragraph of section 3.1.

4. Comments: Why authors used XGBoost justify it

Response: In the process of application XGBoost, we obtained 88.3% good accuracy for machine learning classification tasks with 5 classes, which is close to 19% higher than our baseline model, so we think this result is acceptable.

5. Comments: Proofread the entire manuscript

Response: We have proofread the full text and corrected some grammatical errors.

6. Comments: Authors follow the recently published paper “Abdulkadhar, S., Murugesan, G., & Natarajan, J. (2020). Classifying protein-protein interaction articles from biomedical literature using many relevant features and context-free grammar. Journal of King Saud University-Computer and Information Sciences, 32(5), 553-560.”

Response: We have already referred to the paper you mentioned and cited it in the introduction section.

---

## [Decision Letter · Decision Letter 1]

20 Mar 2023

User Review Analysis of Dating Apps based on Text Mining

PONE-D-22-30174R1

Dear Dr. Chen,

We’re pleased to inform you that your manuscript has been judged scientifically suitable for publication and will be formally accepted for publication once it meets all outstanding technical requirements.

Kind regards,

Viacheslav Kovtun, Dr.Sc., Ph.D.

Academic Editor

PLOS ONE

Additional Editor Comments (optional):

Reviewers' comments:

Reviewer's Responses to Questions

**Comments to the Author**

1. If the authors have adequately addressed your comments raised in a previous round of review and you feel that this manuscript is now acceptable for publication, you may indicate that here to bypass the “Comments to the Author” section, enter your conflict of interest statement in the “Confidential to Editor” section, and submit your "Accept" recommendation.

Reviewer #1: All comments have been addressed

2. Is the manuscript technically sound, and do the data support the conclusions?

Reviewer #1: Partly

3. Has the statistical analysis been performed appropriately and rigorously? 

Reviewer #1: Yes

4. Have the authors made all data underlying the findings in their manuscript fully available?

Reviewer #1: Yes

5. Is the manuscript presented in an intelligible fashion and written in standard English?

Reviewer #1: Yes

6. Review Comments to the Author

Reviewer #1: After giving the piece my complete attention, I can say that the authors have addressed each and every one of my concerns.

7. PLOS authors have the option to publish the peer review history of their article (what does this mean?). If published, this will include your full peer review and any attached files.

Reviewer #1: No

---

## [Editor Report · Acceptance letter]

30 Mar 2023

PONE-D-22-30174R1 

User Review Analysis of Dating Apps based on Text Mining 

Dear Dr. Chen:

I'm pleased to inform you that your manuscript has been deemed suitable for publication in PLOS ONE. Congratulations! Your manuscript is now with our production department. 

Kind regards, 

on behalf of

Professor Viacheslav Kovtun 

Academic Editor

PLOS ONE